# Current Scenario of Exogenously Induced RNAi for Lepidopteran Agricultural Pest Control: From dsRNA Design to Topical Application

**DOI:** 10.3390/ijms232415836

**Published:** 2022-12-13

**Authors:** Vívian S. Lucena-Leandro, Emanuel F. A. Abreu, Leonardo A. Vidal, Caroline R. Torres, Camila I. C. V. F. Junqueira, Juliana Dantas, Érika V. S. Albuquerque

**Affiliations:** 1Embrapa Recursos Genéticos e Biotecnologia, Brasília 70770-917, DF, Brazil; 2Department of Cellular Biology, Institute of Biological Sciences, Campus Darcy Ribeiro, Universidade de Brasília—UnB, Brasília 70910-9002, DF, Brazil; 3Department of Agronomy and Veterinary Medicine, Campus Darcy Ribeiro, Universidade de Brasília—UnB, Brasília 70910-9002, DF, Brazil

**Keywords:** biopesticide, gene target, genome, insect, silencing, topical, validation

## Abstract

Invasive insects cost the global economy around USD 70 billion per year. Moreover, increasing agricultural insect pests raise concerns about global food security constraining and infestation rising after climate changes. Current agricultural pest management largely relies on plant breeding—with or without transgenes—and chemical pesticides. Both approaches face serious technological obsolescence in the field due to plant resistance breakdown or development of insecticide resistance. The need for new modes of action (MoA) for managing crop health is growing each year, driven by market demands to reduce economic losses and by consumer demand for phytosanitary measures. The disabling of pest genes through sequence-specific expression silencing is a promising tool in the development of environmentally-friendly and safe biopesticides. The specificity conferred by long dsRNA-base solutions helps minimize effects on off-target genes in the insect pest genome and the target gene in non-target organisms (NTOs). In this review, we summarize the status of gene silencing by RNA interference (RNAi) for agricultural control. More specifically, we focus on the engineering, development and application of gene silencing to control Lepidoptera through non-transforming dsRNA technologies. Despite some delivery and stability drawbacks of topical applications, we reviewed works showing convincing proof-of-concept results that point to innovative solutions. Considerations about the regulation of the ongoing research on dsRNA-based pesticides to produce commercialized products for exogenous application are discussed. Academic and industry initiatives have revealed a worthy effort to control Lepidoptera pests with this new mode of action, which provides more sustainable and reliable technologies for field management. New data on the genomics of this taxon may contribute to a future customized target gene portfolio. As a case study, we illustrate how dsRNA and associated methodologies could be applied to control an important lepidopteran coffee pest.

## 1. Introduction

Insects play an important role in global crop loss, whether feeding on plants, acting as vectors of other diseases, or both [1]. It is estimated that these arthropods are responsible for reducing world food production by 20% [2]. They also reduce household food security at the post-harvest level [3]. Adaptive interventions are required, otherwise this damage may increase within the climate change scenario [4].

Conventional pesticides are currently used to control agricultural insect pests. However, resistant pest populations are frequently selected for by continuous exposure to a given molecule (e.g., chlorantraniliprole) [5]. This context has prompted research on alternative ways to mitigate the negative impacts of insects on the field, which is favored by the increasing knowledge of plant–pest interactions. Pest management strategies that utilize novel modes of action (MoA), such as gene expression silencing in response to RNA interference (RNAi), may bypass pesticide resistance and avoid chemical pesticides [6].

Since the first report involving the expression knockout of two target genes in the nematode *Caenorhabditis elegans* by antisense RNA molecules [7], the RNAi pathway has been reported as highly conserved in almost all eukaryotes [8] and described as a fine-tuned form of gene regulation [9,10] and defensive barrier [11,12,13]. Additionally, RNA silencing contributes to suppression of transposable elements [14], DNA elimination [15], heterochromatin formation [16] and posttranscriptional repression of cellular genes [17].

The RNAi gene silencing pathways are highly conserved in insects and operate basically in three distinct ways, according to the kind of small RNA responsible for triggering the silencing effect: siRNAsin the siRNA pathway for endo/exogenous dsRNAs; microRNAs (miRNAs) in the miRNA pathway [18]; and P-element induced wimpy testis (Piwi)-interacting RNAs (piRNAs) in the piRNA pathway [19,20].

In arthropods, the first step in triggering the dsRNA-mediated RNAi pathway is the uptake of dsRNA molecules from the external environment, which depends on transmembrane channel-mediated and/or endocytosis-mediated mechanisms [21], followed by interaction of dsRNA with degradation machinery located inside the cells. Several proteins implicated in dsRNA uptake have been studied and described. In the model organism *C. elegans*, four proteins belonging to the *Sid* family (SID-1, SID-2, SID-3 and SID-5) were related to dsRNA uptake efficiency (reviewed by [22]). SID-1 is a channel protein that binds to the dsRNA molecules required for systemic interference [23]. SID-1 orthologs have been reported in lepidopteran pests [24,25]. Clathrin belongs to another class of proteins which promotes clathrin-dependent endocytosis [26]. In *Acyrthosiphon pisum*, the pea aphid, genes involved in the clathrin-dependent pathway were induced 12 h after feeding with dsRNA [27]. Pattern recognition receptors (PRRs) interact with membrane receptors, performing an important function in the uptake of dsRNA through the endocytosis pathway [28,29]. In this context, PRRs in the lepidopteran pests *Helicoverpa armigera*, *Plutella xylostella* and *Spodoptera exigua* were reviewed [30].

Once in contact with insect cellular membranes, dsRNA molecules are taken up by endocytosis [31]. In the cytoplasm environment, dsRNA attaches to a complex formed by the Dicer-2 ribo-nuclease (DCR2), dsRNA binding protein (R2D2) and other associated proteins like pre-mRNA splicing factor (SMD1), Arsenite Resistence Protein 2 (ARS2) and Nuclear Cap-Biding Complex (CBC). The interaction of dsRNA molecules with DCR2, R2D2, SMD1, ARS2 and CBC promote a conformation capable of recruiting the Argonaute protein (AGO) to assemble of a second complex, called RISC (RNA-induced silencing complex). At the mature RISC complex, part of the dsRNA sequence is used as a guide/bait to identify the target mRNA, degrade it and avoid its expression.

In terms of ability to spread the dsRNA signal, two types of events are described: the cell-autonomous route, in which a single cell undergoes the effect of dsRNA presence, and the non-cell-autonomous route, where the interfering effects travel toward tissues/cells/organs distinct from the initial point of application or production. The non-cell-autonomous route is indicated for the development of new MoA assets in RNAi-based pest control. Further details on dsRNA uptake and the progress of this knowledge over time in different species has been reviewed elsewhere [22,32].

Gene silencing is a promising technology that aims to contribute to the control management of several insects of agronomic interest, especially in its topical version, termed spray-induced gene silencing (SIGS) [25]. Compared to conventional pesticides, the SIGS approach has the advantage of high specificity towards the target organism and fast environmental degradation into innocuous compounds [33,34]. Therefore, this novel pest management procedure has the potential to reduce the employment of conventional insecticides, with intrinsic advantages related to the regulatory restrictions inherent to genetically modified organisms (GMOs), given that dsRNAs can be delivered via topical methods while still maintaining target specificity [35]. In this paper, we highlight the siRNA pathway for exogenous dsRNA, which may represent the next phase of species-specific pest management [36].

## 2. dsRNA-Mediated Silencing to Control Insects

### 2.1. dsRNA-Mediated Silencing Core Machinery

In insects, the double-stranded RNA (dsRNA)-mediated gene silencing technique was initially used in a study on *Drosophila melanogaster* functional genomics [37]. As other insects were studied and their mechanism of gene silencing via RNAi was consequently better understood, it was clear that RNAi efficiency varied among different insect families depending on the insect’s ability to trigger the gene silencing machinery. This RNAi efficiency relies on cellular uptake, dsRNA degradation, inter- and intracellular transport, and processing of dsRNA into short/small interfering RNA (siRNA) [18].

Upon uptake of dsRNA by the target insect, the first step for intracellular siRNA—the mediated pathway, in general—is the cleavage of the exogenous dsRNA by a specific endonuclease from the *Dcr2* family [18]. The fragments generated by DCR2 are used as a template by R2D2, an RNA-dependent RNA polymerase, leading to amplification of the silencing stimulus. Next, the siRNA molecules produced by the DCR2/R2D2 complex guide the activity of an RNA-induced silencing complex (RISC), formed by Argonaute 2 (AGO2) and some associated proteins, to conduct the homologous RNA degradation [38]. There is a positive co-relationship between RNAi efficiency and the core RNAi pathway gene expression pattern. Therefore, the next requirement for a successful RNAi-based strategy is to understand how the silencing machinery operates in the target insect.

The main elements of the siRNA pathway studied to elucidate the dsRNA processing are DCR2, R2D2, and AGO2. In the case of the fall armyworm (*Spodoptera frugiperda*; Lepidoptera), the basic transcription levels of these core elements of RNAi machinery are similar to those observed in western corn rootworm (*Diabrotica virgifera*; Coleoptera) and southern green stink bug (*Nezara viridula*; Hemiptera) [39]. All three taxa are considered highly efficient species on which to employ RNAi strategies [40,41,42], from phylogenetic orders whose response to dsRNAi varies.

The impact of dsRNA delivery via injection or feeding on the modulation of the RNAi core genes remains unclear. This was investigated in the European corn borer (ECB) (*Ostrinia nubilalis*; Lepidoptera), a low RNAi efficient species [6]. The study revealed only one transcript for each of the three core RNAi pathway genes (*Dcr2*, *R2D2* and *Ago2*) denominated *OnDcr2*, *OnR2D2* and *OnAgo2*. Expression levels of these genes after dsRNA injection remained steady during the injection assay, while *OnDcr2* alone was upregulated upon an artificial diet assay. Furthermore, analysis of the predicted domains provided functional information concerning conformational differences of *OnAgo2*, *OnR2D2* and *OnDcr2* that could justify the low efficiency of the RNAi apparatus in ECB [6].

### 2.2. dsRNA Designing

Insects present different levels of susceptibility to dsRNA. Some orders, such as Coleoptera, perform strongly, while others, including Lepidoptera and Diptera, exhibit highly variable outcomes in response to dsRNA treatment, requiring detailed tactics to deliver better-performing solutions [43].

The employment of RNAi for pest control purposes has been improved with advances in high-throughput sequencing (genomic and transcriptomic) and bioinformatic tools [44]. Deeper molecular information on the target insects has provided high specificity to the identification of essential genes for which to create silencing molecules. Hence, large datasets enable strict homology levels between the dsRNA and its corresponding mRNA target. dsRNA design for commercial purposes starts with the development of a preliminary pipeline, where software settings must be adjusted for each specific trait vs. organism analyzed [45,46,47]. Then, minimum off-target effects criteria are set to select functional siRNA sequences that guarantee near-perfect matching [48]. Accordingly, carefully designed dsRNA makes it possible to obtain broad-spectrum or extremely specific molecules targeted to the same gene in different insect species, or even within species of the same genus [49].

The precaution of excluding cross-kingdom sRNAs (ck-sRNAs) helps avoid undesirable effects on non-target species. For this purpose, when preparing samples for the sequencing step, it is necessary to treat samples in order to avoid contamination with other organisms (plant cells, endogenous microbiota, parasites) [50], determine the number of biological replicates and calculate the correct amount of reads to achieve sufficient genome coverage (typically three biological replicates with 5–10 million reads each). Sequencing of sRNAs is carried out using a wide variety of high-throughput technologies [51,52], including MiSeq and HiSeq (Illumina Inc., San Diego, California, EUA), SMRT PacBio (Pacific BioSciences, Menlo Park, California, EUA), and Roche 454 technologies (Branford, Connecticut), depending on the output range and total reads per run required [53]. These NGS technologies also allow direct sequencing of cDNA produced from messenger RNA (RNA-seq), enabling the de novo construction of the transcriptome without an anchoring genome [54,55]. For instance, RNASeq (Illumina) and digital gene expression tag profile (DGE-tag) were used to screen optimal RNAi targets in Asian corn borer (ACB) (*Ostrinia furnalalis*). Larval-stage specific expression genes were selected for RNAi testing by spraying dsRNA on larvae, reaching mortalities of 73% to 100% at 5 days after treatment [56]. The combination of DGE-tag with RNA-seq is a rapid way to select candidate target genes for RNAi [57].

For siRNA-mediated silencing, high pairing between the target RNA and siRNA is more critical than dsRNA-mediated methods. Furthermore, the siRNA length must be observed for more successful RNAi silencing, considering lepidopteran siRNA populations 20 nt long were observed in some species [58]. BLAST search is an important tool, although not very accurate for short sequences such as siRNAs. Other software to select functional siRNAs, such as PFRED [59] and siRNA-Finder (si-Fi) [60], is available.

### 2.3. Genomic Data on Lepidopteran Pests

Fully sequenced insect pest genomes support assertive gene targeting for dsRNA design [44]. According to the NCBI, the Insecta class (Figure 1a) has 3091 deposited genomes. From these data, 1831 are reference genomes, while 220 were annotated using the NCBI RefSeq.

Lepidopteran insects are holometabolous butterflies and moths whose life cycles comprise the egg, first to fourth instar larvae, pupa and sexually dimorphic adult [61]. According to [62], Lepidoptera is the second most diverse order (157,424 recognized species) and includes the most devastating agricultural pests in the world [63]. For the Lepidoptera order, we found 1611 genomes, being the Ditrysia clade (Figure 1b) represented by 1540 total genomes, 836 of which were reference genomes and 34 annotated by NCBI RefSeq.

The Yponomeutoidea superfamily (Figure 1c) contains 16 deposited genomes, with four reference genomes and the only NCBI RefSeq genome annotated in the Plutellidae family—the well-studied *Plutella xylostella* (Figure 1c), which has genome assembly at the chromosome level. However, other families lack genomic data, such as the cosmopolitan Lyonetiidae family (Figure 1c), which has about 210 described species, encompassing important pests that are usually leaf and branch mining and parasitize dicotyledons [64]. Lyonetids include miner pests which cause wilting and defoliation to: fruit shrubs, such as apple, pear, peach, apricot and cherry in Europe and Asia; and coffee in Americas [65,66,67]; ornamental plants [68,69,70,71]; medicinal plants [72]; trees, such as willow and poplar and bushes [73,74,75]. Unlike other lyonetiid miners, *Leucoptera coffeella* or coffee leaf miner (CLM) feeds exclusively on *Coffea* spp. plants in the Neotropical region [76]. The leaf damage caused by the CLM attack cause productivity losses estimated at 87% and defoliation of up to 75%, depending on the season [77,78]. Recently obtained sequences of *L. coffeella* have generated large genomic, transcriptomic and proteomic information at the molecular level [79], contributing to the development of RNAi research in Lepidoptera by improving the knowledge of the RNAi machinery and the selection of highly specific gene targets.

### 2.4. Insect Target Genes for dsRNA Silencing

To reduce or eliminate the use of chemical pesticides harmful to health and the environment, RNAi has been increasingly developed and tested in agricultural pests. In Table 1, we show some validated genes that are not used in in planta silencing (non-GMOs). Insect gene-silencing papers reported before 2017 and some related to pests occurring in the Neotropical Region has been compiled and reviewed elsewhere [49,80].
ijms-23-15836-t001_Table 1Table 1dsRNA-validated genes for insect silencing by non-transformative methods.YearSpecies (Order)NCBI: txidTargetDeliveryMortalityReference2017*Anthonomus grandis*(Coleoptera)7044*AgraCHS2*microinjection100%[81]2019*Bactrocera dorsalis*(Diptera)27457*Tssk1*artificial diet58.99%[82]*Tektin1*64.49%2019*Myllocerus undecimpustulatus undatus*(Coleoptera)1811735*Prosα2*injection and feeding78.60%[83]*RPS13*64.10%*Snf7*92.70%*V-ATPase A*43.10%2020*Anoplophora glabripennis*(Coleoptera)217634*IAP*artificial diet90%[84]*SNF7*75%*SSK*80%2021*Diaphorina citri*(Hemiptera)121845*DcCP64*soaking72%[85]2021*Leptinotarsa decemlineata*(Coleoptera)7539*PSMB5*leaf soaking50–90%[86]2021*Plautia stali*(Hemiptera)106108*vATPase*injection and feeding100%[87]*IAP**MCO2**Snf7*2022*Brassicogethes aeneus*(Coleoptera)1431903α*COP*leaf soaking62%[88]


The dsRNA to silence the *AgraCHS2* gene resulted in 100% adult mortality when microinjected into the cotton boll weevil (*Anthonomus grandis*; Coleoptera). This gene is essential for peritrophic membrane biosynthesis, intestinal epithelium protection, and nutrient assimilation [81]. Subsequently, the *Tssk1* and *Tekin1* genes (important for male fertility) were silenced, causing the death of 58–64% of *Bactrocera dorsalis* (Diptera) individuals by artificial diet [82].

In the Sri Lanka weevil (*Myllocerus undecimpustulatus undatus*; Coleoptera) silencing was validated through injection and feeding the genes *Prosα2* (proteasome subunit alpha type 2) [89], *RPS13* (structural element of the 40S subunit) [90], *Snf7* (endosomal sorting complex required for transport III- ESCRT-III) [91] and *V-ATPase A* (transmembrane ATP-driven proton pump) [92]. The mortality rate ranged from 43.1% to 92.7%.

The Asian long-horned beetle (ALB), a polyphagous wood-boring species (*Anoplophora glabripennis*; Coleoptera). Bioassays in which larvae were fed with dsRNA resulted in the death of 75–90% of the individuals, upon silencing the inhibitor of apoptosis (*IAP*), *SNF7* and snakeskin (*SSK*) genes [84]. Still, the Asian citrus psyllid (ACP) vector (*Diaphorina citri*; Hemiptera) transmits the Citrus Huanglongbing disease (HLB). Tests with dsRNA to silence the *DcCP64* gene, responsible for the synthesis of the 64-like cuticle protein, resulted in 72% of the psyllids’ deaths [85]. However, the brown-winged green stinkbug (*Plautia stali*; Hemiptera), known for infesting various fruits and crop plants, showed 100% of death when the insects were treated with RNAi directed to the genes *vATPase*, *IAP*, *MCO2* and *Snf7* [87].

The Colorado potato beetle (CPB) *(Leptinotarsa decemlineata*; Coleoptera) was subjected to PSMB5 gene silencing, which is part of the ubiquitin/proteasome machinery, reaching between 50% and 90% mortality depending on the life stage of the CPB. The study ended up generating an RNA-based biopesticide, the Ledprona^®^, which is being reviewed for registration at the United States Environmental Protection Agency (EPA) [93].

The order Lepidoptera contains several highly destructive representatives that generally show low mortality rates when subjected to RNAi-based tests [94]. Despite the limitations of the gene silencing effect in lepidopterans, potential solutions may still exist, as there are possibilities that have not yet been tested [95]. In Table 2**,** we listed works published between 2018–2022 on the validation of lepidopteran pest genes silenced by exogenous application.
ijms-23-15836-t002_Table 2Table 2Lepidopteran target genes validated for dsRNA silencing. The taxonomic positions of these genes’ corresponding species are depicted in Figure 2.YearSpecies (Family)NCBI:txidTargetDeliveryMortalityReference2018*Helicoverpa zea*(Noctuidae)7113*TipE*microinjection12–16%[96]*GluCl*12–16%*Para*12–16%*Notch*12–16%2018*Plutella xylostella*(Plutellidae) 51655*AChE*topical foliar application69–74%[97]2018*Heliothis virescens*(Noctuidae)7102*PBAN*topical foliar applicationand Injection50–60%[98]*Helicoverpa zea* (Noctuidae)711350–60%2021*Hyblaea puera*(Hyblaeidae)268502*HpEcR*topical foliar application46%[99]*HpCHS1*30%*HpChi-h*32%2021*Spodoptera exigua*(Noctuidae)7107GNAFfeeding48%[100]2021*Chilo suppressalis*(Crambidae)168631*ND*topical foliar application50%[101]*GPDH*50%*MSL3*50%2021*Tuta absoluta*(Gelechiidae)702717*v-ATPase B*topical foliar application70%[102]*JHBP*70%


The *AChE* gene was silenced via soaking in diamond back moth (*Plutella xylostella*) one of the main pests of cruciferous vegetables. This gene is responsible for the synthesis of acetylcholine esterase—the primary target of commercial insecticides—which interrupts the action of neurotransmitters. To increase stability, researchers tested a concatemerized form of the molecule. The mortality rate was up to 72%, higher than that observed with the non-concatemerized control [97].

The corn caterpillar (*Helicoverpa zea*) was subjected to tests involving RNAi using genes that are targets of commercial insecticides: *Para* (paralytic effect), *TipE* (temperature-induced paralysis), *GluCl* (glutamate chloride channel), and *Notch* (encodes proteins that make up neuronal cells). Three delivery methods were used: microinjection, egg immersion, and larval feeding. Microinjection of eggs of the *GluCl*, *Para*, and *TipE* genes reduced hatching rates, while the *Notch* gene showed no difference. None of the genes were effective for larval feeding and egg immersion methods [96].

Pinworm (*Tuta absoluta*) is the most aggressive tomato pest in South America, Africa and Asia. This lepidopteran, which feeds on mesophyll, was subjected to tests with RNAi of the genes: *v-ATPase B* (keeps the midgut lumen alkaline by increasing amino acid absorption) and *JHBP* (essential for development and reproductive maturation). Topical application on the leaf surface resulted in 70% of larval mortality for the two target genes [102].

The Asiatic rice borer (*Chilo supressalis*) is one of the world’s main crop pests. As its chemical control is expensive, the RNAi technique has become a possible option for the borer’s management. Three selected genes—*ND* (NADH dehydrogenase), *GPDH* (glycerol 3-phosphate dehydrogenase), and *MSL3* (male specific lethal 3)—were tested. Rice leaves were brushed with bacterial dsRNA solutions that contained newly hatched larvae. The insect mortality rate was 50% for each gene [101].

Corn caterpillar (*H. zea*) and tobacco caterpillar (*Heliothis virescens*) were subjected to control tests containing dsRNA of the *PBAN* target gene. dsRNA delivery by artificial larval diet or pupae injection caused a mortality rate that ranged from 30 to 60%. A delay in larval development and some interference in the development of pupae were also observed in the two agricultural pests analyzed [98].

The teak defoliator (*Hyblaea puera*) causes severe defoliation to teak, a tree of paramount commercial importance in forestry. Sequences from the *HpCHS1*, *HpChi-h*, and *HpEcR* genes, related to chitin metabolism, were used in the construction of dsRNAs that were administered to the larvae via topical application on the leaves. It was found that 30–46% of the treated larvae died; a large number of deformed pupae were also observed, in addition to the deformed pupae [99].

Assuming that gene silencing can occur in Lepidoptera species, we believe that the CLM (*Leucoptera coffeella*) can also be controlled by exogenous dsRNA technologies. Aiming to develop biopesticide solutions to this important coffee pest, our research group has performed the full genome PACBio and paired-end Illumina combined DNA sequencing from pupae samples. The generated data allowed for nuclear genome, transcriptome and proteome analyses as a basis for the discovery of RNAi mechanisms specific to the Lyonetiidae family, as well as the selection of target genes. This information is essential to RNAi development, as the taxonomically closest complete genome is in the very distant Yponomeutoidea taxon [79].

## 3. Validation Tests of the dsRNA Candidates for Proof of Concept

After synthesizing the dsRNA molecules in the laboratory, validation in a controlled rearing environment is required before field application, either by direct delivery to the insect or indirect application on the plant for subsequent ingestion. About general assay aspects, an interesting study provided important insights into how to apply dsRNA exogenously to plants in *Arabidopsis thaliana* [103]. Under different physiological conditions (plant age, time of day, soil moisture, high salinity and heat and cold stress) and via various application media (brush spreading, spraying, infiltration, inoculation, needle injection and pipetting), the best results were associated to application at night, low moisture soil, brush spreading, spraying and pipetting.

Aiming to further test the small-sized *Leucoptera coffeella* larvae (1-4 mm in length), we found some useful techniques that could be applied to testing larvae and pupae in vitro or ex vitro. In this way, we discuss some dsRNA application techniques used in other insects that can be tested according to our reality, considering the size and development stages of *L. coffeella* and the characteristics of the coffee plant.

To insects depending on the living plant tissue to complete the life cycle, as *L. leucoptera*, the plant infiltration seems to be very suitable. This procedure is performed by syringe without a needle, pressing the dsRNA-containing solution under the abaxial face of the leaf. It may be a viable method for validating the dsRNA in plants, as it is possible to visualize the entry of the solution into the leaf. Sometime later, the liquid diffuses inside the plant tissues and is no longer visible. The infiltration process is carried out in sunflowers [104]. Even if it is an easy method, it is quite variable in effectiveness, depending on plant anatomy.

Microinjection is an option proved to work even to 1 mm insect developmental stages. Tests performed with dsRNA in the cotton boll weevil (*Anthonomus grandis*) showed a 93% reduction in oviposition and 100% of adults death to 1mm microinjected larvae [81]. With *H. zea* 1mm diameter eggs, microinjection resulted in 12–16% mortality of larvae after silencing five genes [96]. Nine dsRNA molecules specific to *Nezara viridula* (southern green stink bug) showed an average mortality rate of 90% after microinjection into 3.2mm nymphs [105,106].

Immature stages could also be tested by detached leaves, as the 48% mortality reported to the beet armyworm (*Spodoptera exigua*) (Table 2) fed with dsRNA treated leaves [100]. Also, worth of testing is the soaking method used in *Diaphorina citri* (Asian citrus psyllid) nymphs [85,107].

## 4. dsRNA-Based Products

### 4.1. Formulation with Nanocarriers

Once the target genes that impair insect proliferation are chosen and preliminary assays are realized in laboratory and greenhouse conditions, other constraints inherent to the dsRNAi approach should be addressed. The stability of dsRNA molecules after field application and in insect gut conditions is crucial to extending the material’s half-life length.

It is known that to control lepidopteran pests via dsRNA-based products, avoiding dsRNA degradation by RNases and high pH levels is a sine qua non situation [108]. After application to the plants, the dsRNA molecule must get to the target gene inside the insect cells [109]. The primary constraint of using molecules ingested by insects is their degradation in the gut. Most of the dsRNases present in lepidopterans’ guts are basophilic, presenting an optimum pH of around 9.0 [26]. dsRNA stability is influenced by high pH values due to its chemical structure. In alkaline conditions, dsRNA can be hydrolyzed in some regions [26]. Noticeably, the drawbacks in Lepidoptera are even more significant [110] because pH in the lepidopteran gut can reach 12 [111]. This factor is probably the biggest challenge to making dsRNA applicable to lepidopteran insect control [110]. Beyond the gut, dsRNases present high activity in other body fluids such as saliva and hemolymph, which cause the degradation before the dsRNA can been processed by the insect [23].

The fate of dsRNA is short both in soil and aquatic environments [34,112]. Microbial nucleases present in the soil and on leaves, UV-radiation and run-off due to dew and rain can significantly limit the availability of dsRNA to the pest [112,113].

Fortunately, most of those problems have been gradually overcome with improvements and recent findings in chemistry and nanotechnology. Thus, dsRNA formulations containing protection agents, surfactants, and diluents may be required [114]. Many nanomaterials are used to form complexes with dsRNA, as presented below—especially the cationic ones that present positive extremities, able to bond with the negative parts of dsRNA molecules.

### 4.2. Peptides

The peptide transduction domain (PTD) may be fused to dsRNA binding domain (DRDB) molecules, allowing a more efficient internalization by the cell and consequently enhancing gene silencing in *Anthonomus grandis*. Although the PTD-DRBD complex made the dsRNA more stable and efficient in the presence of nucleases in the *A. grandis* gut [115], the dsRNA was harder to synthesize. An alternative is using branched amphiphilic peptide capsules (BAPCs), which are more easily synthesized than PTD, to protect dsRNAs [116]. These nano capsules are made of natural amino acids, and are water soluble and resistant to detergents, proteases and chaotropic agents. The association of BAPCs to dsRNA led to premature deaths of pea aphid (*Acyrthosiphon pisum*) when fed with a diet containing BAPCs-dsRNA complex [117].

### 4.3. Transfection Reagent

Lipofectamine^®^ 2000 (Reagent Catalog number: 11668030, Thermo Fisher Scientific, Life Sciences Solutions, 5781 Van Allen Way, Carlsbad, CA, USA 92008, Invitrogen) is a transfection reagent able to inhibit nuclease activity and characterized by the presence of cations at the phospholipid bilayer. These characteristics allow lipofectamine to overcome the repulsion relation between cell membranes and nucleic acids, becoming a good candidate to be added to dsRNA formulations [116]. When used to treat the brown stink bug (*Euschistus heros*; Hemiptera), lipofectamine was able to increase the mortality of second-instar nymphs by 15% after 14 days of artificial feeding, in comparison with the naked dsRNA molecules, which caused 33% mortality [118].

Another transfection reagent, Cellfectin^®^ II Reagent (CFII) (Catalog number: 10362100, Thermo Fisher Scientific, Life Sciences Solutions, 5781 Van Allen Way, Carlsbad, CA, USA 92008), was tested complexed with dsRNA against *Spodoptera frugiperda*. One group of larvae was fed with CFII-dsRNA complex and the other group received naked dsRNA. The diet without CFII caused 25% mortality, while the group that received CFII-dsRNA registered 55%. These results indicate that CFII was able to protect dsRNA molecules from the high pH and the nucleases of the hemolymph and midgut lumen, increasing dsRNA’s final efficiency [119].

### 4.4. Macromolecular Polymers

Star polycation (SPc) is a cationic amino acid dendrimer nanocarrier that is able to enhance gene transfection efficiency. Its dsRNA association capability, due to its positive charge, allowed RNAi silencing of the *CYP6CY3* gene up to 84.3% mortality in *Aphis gossypii* [120]. SPc was also used in association with dsRNA to verify its capability in the control an polyphage aphid (*Myzus persicae*; Hemiptera) [121]. To avoid water repellency a 0.1% detergent solution was added to the formulation. The penetration efficiency at the aphid, measured by fluorescence tests, showed that with SPc, dsRNA molecules were able to penetrate the aphid’s whole body. The *vestigial* (*vg*) and *Ultrabithorax* (*Ubx*) genes were downregulated 44.0% and 36.5%, respectively, 24 h after treatment.

SPc polymerized with DMAEMA (2-N-(dimethyl aminoethyl) methacrylate) was made in association with dsRNA to control *Chilo suppressalis* by artificial feeding. The SPc protection showed an increase of 60% in larvae mortality [122]. In *S. frugiperda*, the SPc complexed with dsRNA was tested in Sf9 cells, hemolymph and midgut lumen contents collected from *S. frugiperda* larvae. The performance of the complex and naked molecules was measured by UV tests at 488 nm, showing that SPc was able to enhance dsRNA molecules’ stability. After 12 h of incubation, the UV signal was much stronger in cells that absorbed the complex dsRNA/SPc than in naked dsRNA, showing that SPc can promote dsRNA uptake by cells. Also, SPc was able to protect dsRNA molecules from being degraded by RNase A and the insect hemolymph [123].

Polymers containing guanidine were developed to stabilize dsRNA molecules in very alkaline environments and to protect these molecules from nucleases. Since dsRNA molecules are very sensitive to high pH environments because of the hydrolyzation caused to double-stranded molecules [26], it is necessary to prevent degradation in lepidopterans’ guts, which can reach pH 12 [111]. dsRNA can form complexes with the guanylate polymer due to its positive charge. In laboratory ex vivo tests, a guanylate polymer was able to avoid dsRNA degradation for over 30 h at pH 11, while the naked molecules were degraded after 10 min in *Spodoptera exigua* gut juice. In vivo bioassays showed that the mortality associated with the dsRNA complexed with the guanylate polymer was around 53.3%, while the naked molecules caused 16.7% of mortality [124].

Chitin can be deacetylated and become a natural material called chitosan. This cationic nanopolymer is composed of biodegradable and biocompatible molecules [125]. Chitosan nanoparticles (CNPs) were complexed to dsRNA molecules and sprayed over chickpea leaves to control the lepidopteran *Helicoverpa armigera*. The CNPs/dsRNA complex led to a reduction in length and weight of the larvae when compared to naked-molecule silencing [126]. Chitosan (CS) can be further improved by other materials, such as sodium tripolyphosphate (STPP), a nanosized cross-linker able to increase the protection of the chitosan-dsRNA complex, forming a CS-STPP-dsRNA complex [127].

### 4.5. Other Materials

Layered double hydroxide (LDH) clay nanosheets have a flat, hexagonal, positively charged structure that interacts with dsRNA molecules, forming a complex named BioClay. This structure allows for dsRNA detection on the surface of leaves even 30 days after application via foliar spray. Moreover, it also enables improved permanence of the molecules over the leaves, avoiding removal by the application of other products [128].

Carbon Quantum Dots (CQD) are another possible formulation reagent. When associated with dsRNA molecules, CQD was able to improve gene depression by 41% in the gut, 45% in other tissues and 43% in the whole organism. In contrast, control naked dsRNA did not show any expression level reduction. The downregulation caused by silencing resulted in 70% mortality 6 days after diet feeding in *C. suppressalis*. The CQD also showed a very good stability result in comparison to chitosan, Lipofectamine^®^ 2000 and naked molecules of dsRNA when tested on midgut homogenates [129].

## 5. dsRNA Production

In the last few years, dsRNA production methods have been continuously optimized to promote the wider application of this technology. dsRNA can be produced using either in vitro transcription or in vivo expression in bacteria, yeast, microalgae and other species [130,131,132,133,134]. Bacteria are a low-cost alternative for the production of large amounts of dsRNA [135]. *Escherichia coli* HT115 (DE3) were first used as an heterologous dsRNA production system by [136]. Since then, it has become a useful tool for functional studies in invertebrate physiology [134,137,138,139]. Still, other kinds of bacteria, such as *Bacillus thuringiensis* (Bt) [130,133,140], *B. subtilis* [141,142], symbiotic bacteria [143,144,145,146,147], *Pseudomonas syringae*, *Corynebacterium glutamicum* and *Chlamydomonas reinhardtii* are useful biopesticide production strains that have been widely used for the control of several pests.

In addition, eukaryotic species *Saccharomyces cerevisiae* has also been used as a relevant dsRNA production system. *S. cerevisiae* does not contain the core genes *Dicer2* and *Argonaute-2* of the RNAi pathway [132], which allows efficient dsRNA synthesis in *S. cerevisiae* compared with *E. coli* and other bacterial species [148]. Fungi [131] and viruses [149,150] have also been engineered to produce dsRNA.

In experimental settings, three main approaches are used to produce dsRNA: chemical synthesis of NTPs, in vitro synthesis through RNA-dependent RNA polymerases and fermentation through microorganisms. To this goal, in vitro synthesis is best suited for the synthesis of short dsRNAs [151] and to reach high-purity dsRNA standards, despite its relatively high production cost [152].

For field application, the main requirement is the establishment of large-scale production at low cost. Bacterial systems are the most used alternative [138] and promising methods to biologically synthesize long dsRNAs for the control of Lepidoptera species and other insect pests [42,153]. Although in vivo production supplies low-cost dsRNA in high yields, this strategy requires later purification of the product and inactivation of the engineered microbial strain [154,155]. Efficient production involves the proper release of dsRNA from the cells, without affecting dsRNA integrity during the extraction process Methods for the Cost-Effective Production of Bacteria-Derived Double [138]. Despite the risk of compromised integrity inherent to this method, dsRNA extracted from bacteria efficiently induced knockdowns in a lepidopteran cell line [138].

In 2008, the cost of synthetic dsRNA was approximately 12,000 USD/g, dropping to 60 USD/g in 2018 [156]. RNAGri had the ability to produce tons of dsRNA at a cost of 1 USD/g, while Greenlight’s GreenWorX™ system can further reduce the cost of dsRNA synthesis to < 0.5 USD/g [49,152,157,158,159]. In 2009, a study showed that producing 30 mg of dsRNA in vivo was approximately one-third the cost of the in vitro methodology [160].

## 6. Field Application of dsRNA Assets

### 6.1. Foliar Spray

Foliar application is a very practical and convenient SIGS method that has been reducing in cost over time [161]. Reliable dsRNA market products must present convincing low-cost and high-efficiency characteristics. Foliar spray application fits the synthesis price, as the costs of this method are around 0.5 and 1 USD per gram on a cell-free bioprocessing platform [159], with about 2 to 10 g per hectare required [156,162,163]. The convenience of pulverization is that it is a method already used in most production areas, which represents a shortcut to application on farms.

The most advanced commercial foliar spray product launched is the dsRNA-based Ledprona^®^. Its active ingredient is a dsRNA artificially synthesized and applied via foliar spray. It has been demonstrated to be a very promising product in field trials [86].

Another factor that makes RNAi a difficult strategy to be applied is the uptake of the product when applied via foliar spray. Many leaf surfaces are re-covered, mostly by wax, or have a hydrophobic cuticle, which makes absorption more difficult; the presence of trichomes and stomata density and position also interfere directly with the dsRNA absorption by the plants [161].

Naked dsRNA molecules require some kind of protection or association with other compounds to be used as an active principle to any product, in order to avoid being degraded before they reach their final destination and allow the product to reach its maximum potential [159]. The dsRNA strategy currently faces other drawbacks to application, such as the low pH that many plants present on the leaf surface, which causes dsRNA degradation. Furthermore, UV light can reduce the biological activity of dsRNA. Rain, which is usually an excellent factor for dsRNA application, can become a threat to the stability of dsRNA molecules. These are crucial factors that dsRNA strategies must overcome to reach a good market position [164,165].

Using bacteria to deliver dsRNA presents advantages compared to in vitro dsRNA synthesis [29]. Silencing of five different target genes caused significant larval mortality to the CPB after 12 days feeding on DE3 *E. coli* expressing dsRNAs [166]. RNAi effects were also observed by ingestion of bacteria expressing dsRNA in *Spodoptera exigua* [167,168], *Chilo infuscatellus* [57] and *Tuta absoluta* [169].

Feeding with bacteria expressing a homologous dsRNA was effective to silence an immune gene of the noctuid moth *Spodoptera littoralis* [130], although dsRNA delivery to the hemocoel in lepidopteran insects needs high amounts of dsRNA to achieve significant knockdowns. In field conditions, the quantity required is even higher [42,170].

### 6.2. Trunk Injection

Trunk injection is another application method that can be especially useful to apply in the field, particularly for those crops that are perennial plants and those that present woody trunks, such as citrus, coffee and pine [49]. The low efficiency of dsRNA delivery methods can be circumvented by trunk injection, a promising method that reduces environmental impacts. This process relies on the phloem as a channel where dsRNA may remain stable for a long time due to the presence of sap. Besides being a medium free of RNase, the sap has the advantage of facilitating the injected material’s dissipation throughout the plant [171].

The feasibility of the trunk injection process was demonstrated for the first time in citrus trees and vines that were exposed to a single injection of 2 g of dsRNA diluted in 15 L of water [172]. The delivery was confirmed after the detection of dsRNA in the treated plants. Additionally, the authors demonstrated that two hemipteran insects that fed on sap also absorbed dsRNA, suggesting that this method would be an excellent choice for sucking pests. It is important, however, to note that the trunk injection strategy may be costly, as it requires the massive production of dsRNA and a specific injection apparatus [29].

Some products for injection delivery are currently available: Arbojet^®^ https://arborjet.com/ (accessed on 19 September 2022), which proposes injection by the insertion of a drilling plug [171] and ChemJet^®^ (https://www.chemjet.com.au/ (accessed on 19 September 2022) a rechargeable injector to be inserted within a previously pierced hole 

Depending on the characteristics of the insect pest to be controlled, it is important to evaluate which injection type favors dsRNA delivery to the tissues on which the pest feeds—in the case of the CLM, for example, to the palisade parenchyma of leaf mesophyll. It is also important to evaluate the orifice size and the plug fixation options, since the open orifice becomes an entrance port for pathogens. In addition, there is a risk of generating embolisms in the vessels or compartmentalization of the vascular system after piercing. Despite this, efficient trunk injection for systemic conduction in perennials was demonstrated in apple trees with essential oil emulsions applied through 1 mm-wide and 1 cm-deep perforations [173].

A perennial large crop that could benefit from the trunk injection method is coffee. Similar to the apple tree, coffee is a perennial shrub that could be treated by trunk injection to direct dsRNA to the leaf mesophyll, aiming at CLM larval stage feeding, as illustrated in Figure 2.

## 7. Limitations of RNAi Technology in Lepidoptera

During the last decade, many studies have proven that RNAi technology is efficient in pest control mainly via transgenic plants, feeding, trunk injection, or spraying [56,57,174,175]. Lepidopteran insects, however, present variable RNAi efficiency for several reasons, even though their successful uptake of dsRNA has been evidenced in injection, transfection, or transgenic assays that reported good responses to dsRNA stimuli [29,94]. Studies concerning persistence of dsRNA in the digestive tract of lepidopterans indicate that dsRNA can be quickly degraded by nucleases present in the saliva, hemolymph and gut juice [176,177]. This means that the successful usage of dsRNA in a SIGS approach relies on the protection of the dsRNA from degradation in the field and the insect gut. Formulation technologies can also be used to improve cellular internalization of dsRNA and to protect dsRNA against nucleolytic degradation, hence improving overall delivery to the pest [108,124]. Nevertheless, formulation contents might present a risk on their own and the impact of the formulation itself on the environment and non-target organisms (NTO) should be assessed as well.

Other aspects to consider for commercial application of RNAi are cost-efficiency, safety, the delivery to the site of action in the target organism and adverse effects in NTOs. The exposure of NTO is dependent on several parameters, including application rate, timing of application, application method, number of applications, off-site movement of the dsRNA and stability and persistence of the dsRNA [178,179].

Hence, RNAi efficiency varies greatly among different insect species and the major limitations for efficient RNAi include dsRNA instability, refractory gene targets, low efficiency of dsRNA cellular internalization, deficient core RNAi machinery and impaired systemic spreading of dsRNA, which constrains the application of RNAi-based pest management [123].

## 8. Final Considerations

RNAi-based biopesticides have fewer harmful effects than most conventional chemical pesticides and no pest resistance development is expected in the field when using long dsRNA strategies. These molecules are processed in many different siRNAs to silence the target genes, minimizing the probability of acquired RNAi-resistance. Moreover, non-transformative strategies likely prevent insects from adapting and circumventing resistance to dsRNA silencing, perhaps because the intermittent presence of the dsRNAs—in contrast to a constant supply provided by HIGS—presents lower selection pressure [180].

RNAi-based pesticide control of lepidopteran pests encompasses environmental stability and low silencing efficiency limitations that may be surmounted by nanocarriers and adjuvants associated to dsRNAs [181].

From a regulatory perspective, exogenous delivery methods are more likely to be accepted for commercialization due to their biosafety appeal. These products offer pest gene control without introducing GMO plants into the environment. Moreover, the topical application of dsRNAs is expected to be minimally impactful due to their fast environmental fate, low non-target and off-target risks and other advantages over HIGS solutions. As the importance of research and commercial interests in exogenous RNAi technology has risen strikingly in recent years, many forums for discussing the regulatory frameworks for pesticide authorization in the United States (US) and European Union (EU) have formed within the European Food Safety Authority (EFSA), the Organization for Economic Co-operation and Development (OECD) European Co-operation in Science and Technology (COST) and the scientific community [44]. Currently, while dsRNA-based insecticides are being generated, documents from these initiatives are gathering knowledge and outlining the classification and authorization procedures for the environmental risk assessment (ERA) guidelines to this new MoA, considering factors such as the introduction and mobility of dsRNA within target species, its environmental fate, prediction and determination of off-target and non-target effects and resistance development [44,181].

Considering all the aspects discussed above, one can infer that in the medium-term future the control of lepidopterans by topical dsRNA will be widely adopted in world agriculture.

## Figures and Tables

**Figure 1 ijms-23-15836-f001:**
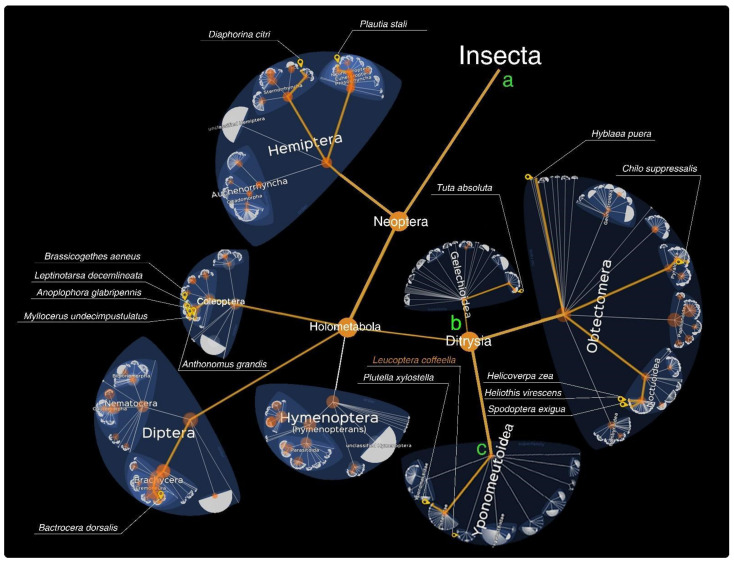
Main taxonomic levels are depicted in a tree, modified from https://lifemap-ncbi.univ-lyon1.fr/ (accessed on 21 September 2022), showing the Insecta class (**a**), the Ditrysia clade (**b**) and the Yponomeutoidea superfamily (**c**). Orange spheres highlight the levels with fully sequenced genome data. Insect species listed in Table 1 and Table 2 (written in white) are marked by a yellow tag. The *L. coffeella* in the Lyonetiidae family is written in orange.

**Figure 2 ijms-23-15836-f002:**
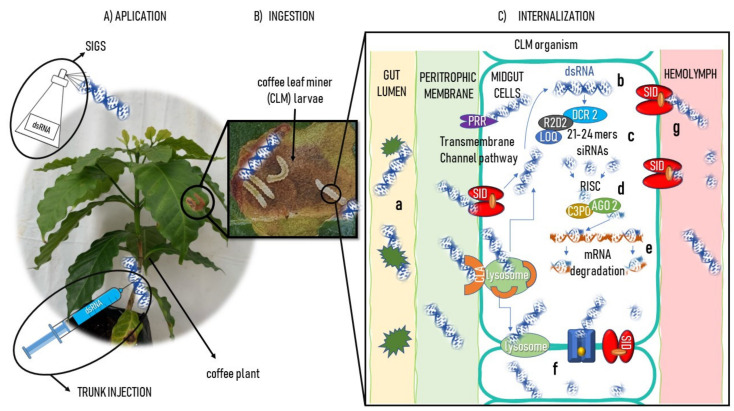
Schematic strategy of RNAi to control *L. coffeella* by long dsRNA-induced silencing of coffee plants. (**A**) dsRNA delivery by SIGS or trunk injection; (**B**) dsRNA oral ingestion by larva feeding the leaf parenchyma; (**C**) dsRNA internalization in the larva: **a** transfer from insect gut lumen to midgut cells by the clathrin-mediated endocytic pathway (CLA), pattern recognition receptors (PRRs), and/or RNA-binding proteins (RBPs); **b** long dsRNA interaction with an R2D2-DCR2 complex associated with LOQ; **c** LOQ recognition of exogenously delivered dsRNA to slicing by Dicer (*DCR2*) with *R2D2* to produce siRNAs; **d** guide-strand selection by AGO2 complexed with C3PO; **e** siRNA-guided silencing by attachment to the target mRNA passenger strand; systemic amplification of the RNAi silencing by dsRNA and siRNA; **f** dispersion through other gut cells by the cytoplasm and **g** diffusion to the hemolymph.

## Data Availability

Not applicable.

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
