# Peer review of "Current Scenario of Exogenously Induced RNAi for Lepidopteran Agricultural Pest Control: From dsRNA Design to Topical Application"

_ijms, 2022, doi:10.3390/ijms232415836_

Round 1
Reviewer 1 Report
This paper provides a well-structured review on application of RNAi technology for insect pest control, with an emphasis to the Lepidoptera pests, taking into account mechanisms of action, as well as production and application concerns. The attached version of the manuscript contains yellow-marked parts that should be corrected and reformulated to make certain parts of the manuscript more readable and clear. Some specific remarks are given below:
OnDcr2, 128 OnR2D2, and OnAgo2 should be explained in 2.1.
Subtitle 2.3. should be reformulated to be more clear and precise.
PM abbreviation should be added to 2.4.
The reviewer suggests writing the whole taxonomic identification (genus+species) throughout the manuscript to make it more readable.
Capitalization of the words in the subtitles should be uniform across the manuscript.
PTD-DRBD abbreviation should be explained in 4.1.
DE3 expressing dsRNA of five different target genes caused significant larval mortality to the CPB 12 days after feeding [161]. – this sentence in Chapter 6 should be better explained.
Resolution of Figure 2 should be increased.
The Final considerations part seems unfinished, please add a few more sentences without references pointing out the authors’ conclusions and perspective on further development of this field.

Reviewer 2 Report
Dear Editor –
I have reviewed the manuscript entitled ‘Current scenario of Exogenously Induced RNAi for Lepidopteran Agricultural Pest Control: from dsRNA design to topical application’ for possible review in your journal ‘International Journal of Molecular Sciences’.
Overall, this is a very through review of dsRNA technology, specifically towards Lepidoptera, and I commend the authors for this review.
I have the following major comments before the manuscript could be accepted.
§ It may be good, in the introduction, to give a simple definition of dsRNA/RNAi technology (a history is provided, but not a definition).
§ I do believe the authors spent great care in writing the article. Yet, many sentence constructs are archaic, in bad English, or plainly wrong. I only highlighted some in the abstract and introduction, as it is not my task as a reviewer to correct the language, but I strongly suggest that the article is edited by a native English speaker.
§ The manuscript is a bit long, albeit an interesting read! One may consider cutting some sections that are not needed? (e.g. 4,175-199: the lengthy treatise of pests in the family Lyonetiidae, and maybe the figure 1; 13, 512-525: this seems to be a repeat of what you mentioned before).
§ Organization: The authors start each main section with an unnumbered mini-introduction. This is not needed. Give a subheading for EACH text. This will also reduce some of the repeats. Also, go over the entire organization agai. There is, for example, no use of havinga section called 2.4.1 of there is no 2.4.2 etc!
§ Section 3 in totality is unclear. Section 3.2 is unclear. What does this have to do with the topic of the paper? Explain this is foliar application to deliver the product. Also, Section 3’s title seems to imply the research pathway all the way up to proof-of-concept. Yet, only few (laboratory) technologies are explained? (so adjust the title, please, of that subsection, instead of the subsection).
Other comments are highlighted as sticky notes in the manuscript.
